# PROCEEDINGS A

# Research

electrical engineering, bioenergetics

robotic fish, energetics, fish school

**Author for correspondence:**
Liang Li
e-mail: lli@ab.mpg.de

# Using a robotic platform to study the influence of relative tailbeat phase on the energetic costs of side-by-side swimming in fish

Liang Li[1,2,3], Sridhar Ravi[4], Guangming Xie[5,6] and Iain D. Couzin[1,2,3]

[1]Department of Collective Behaviour, Max Planck Institute of Animal Behavior, Konstanz, Germany
[2]Centre for the Advanced Study of Collective Behaviour, and
[3]Department of Biology, University of Konstanz, Konstanz, Germany
[4]School of Engineering and Information Technology, University of New South Wales - Canberra, Australia
[5]State Key Laboratory for Turbulence and Complex Systems, College of Engineering, and [6]Institute of Ocean Research, Peking University, Beijing, People's Republic of China

(iD) LL, 0000-0002-2447-6295; SR, 0000-0001-7397-9713

A potential benefit of swimming together in coordinated schools is to allow fish to extract energy from vortices shed by their neighbours, thus reducing the costs of locomotion. This hypothesis has been very hard to test in real fish schools, and it has proven very difficult to replicate the complex hydrodynamics at relevant Reynolds numbers using computational simulations. A complementary approach, and the one we adopt here, is to develop and analyse the performance of biomimetic autonomous robotic models that capture the salient kinematics of fish-like swimming, and also interact via hydrodynamic forces. We developed bio-inspired robotic fish which perform sub-carangiform locomotion, and measured the speed and power consumption of robots when swimming in isolation and when swimming side-by-side in pairs. We found that swimming side-by-side confers a substantial increase in both the speed and efficiency of

## PUBLISHING

locomotion of both fish regardless of the relative phase relationship of their body undulations. However, we also find that each individual can slightly increase their own power efficiency if they change relative tailbeat phase by approximately $0.25\pi$ with respect to, and at the energetic expense of, their neighbour. This suggests the possibility of a competitive game-theoretic dynamic between individuals in swimming groups. Our results also demonstrate the potential applicability of our platform, and provide a natural connection between the biology and robotics of collective motion.

## 1. Introduction

Individuals of numerous fish species have been observed swimming closely together in schools [1,2]. For example, schooling bluefin tuna tend to swim with a lateral distance of between 0.3 and 0.9 body lengths (BL) when schooling [3]. Fish likely swim together to reap many benefits of congregation, which include reduced predation risk [4], increased foraging efficiency [5] and improved environmental sensing capabilities [6]. It has also been hypothesized that, from a biophysical standpoint, swimming in schools may benefit individuals energetically, since it could allow them to extract power from the vortices shed by their neighbours [7–16]. However, by virtue of swimming in close proximity to a conspecific, the resulting fluid mechanic interactions between the adjacent fish are likely to result in a different flow structure from that in solitary swimming [17,18]. While a number of previous studies have revealed the capacity of fish to exploit the local flow structure, such as vortices to minimize the energetic cost of swimming (see [19,20] for reviews), it has proved very difficult to assess whether, and if so how, individuals may exploit hydrodynamic interactions when swimming together.

Since fish must often consider more factors than energy saving when swimming [21,22], and since the reduced stress that results from being in close proximity to others reduces oxygen consumption independently of hydrodynamic interactions [23], it has proven very challenging to determine, conclusively, whether, and if so how, fish really save energy by schooling [24,25]. This co-contribution of confounding factors thus may affect swimming efficiency and energetic costs [24,25]. Previous theoretical studies have sought insights into the flow-field structure in schools using a variety of simple physical models to mimic the hydrodynamic interactions between fish: including thin filaments passively undulating in soap flow [26,27] and rigid foils actively flapping in flow [18,28]. They reveal that the hydrodynamic interactions may contribute to drag reduction of the leader [26], energy saving of the follower [27] or higher efficiency for both [18]. However, most of these physical models [15,16,18,26,27] were heavily restrained not allowing for free swimming (except a few computing models [13,29]) and did not consider three-dimensional effects of the body platform; furthermore, power cost and swimming efficiency were not directly measured. Recent studies show an estimation of the power costs with robotic fish models swimming in a flow tank [16,30], but in that work the movement of the robots was controlled by external fixtures, and thus the costs and benefits of free swimming robots remain unexplored.

In this study, we sought to evaluate the process by which fish may benefit from hydrodynamic interactions when swimming together using biomimetic robots from which we could directly measure power consummation and thus estimate locomotion efficiency. To do so, we introduced a freely swimming high-fidelity sub-carangiform robotic fish platform for exploring hydrodynamic interactions among fish. The locomotion of our robot was controlled by a bioinspired, central pattern generator (CPG) neural network [31]. Our robot fish mimics the kinematics of real fish and is capable of stable self-propelled swimming [16].

Since swimming in pairs is the most common configuration exhibited by schooling fish [12], and as a proof of concept, here we consider the consequences of hydrodynamic interactions between a pair of fish-like robots swimming side-by-side. The influence of the hydrodynamic interactions is evaluated directly using power cost (watts) and swimming speed (metres/second),

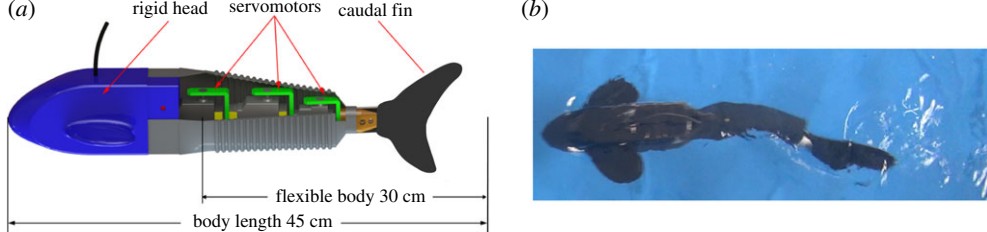

**Figure 1.** The prototype of our robotic fish. (*a*) Inner structure of the robot, which contains a rigid head, a flexible body and a caudal fin. (*b*) A snapshot of our robotic fish swimming by self-propulsion. (Online version in colour.)

which can be compared directly with the cost and speed when swimming alone. We estimate efficiency as how far the robot can swim (the locomotive power) with 1 J energy input within 1 s (the total power), which is similar to the definition of 'miles per gallon' suggested by Schultz & Webb [32]. We also explore here the relationship between the relative undulation pattern (phase difference) exhibited by robots swimming in pairs and the costs and benefits associated with each individual, and verify the effectiveness of our robotic fish model for exploring hydrodynamics in fish schools.

## 2. The platform design

### (a) The robotic fish prototype

Our robotic fish (figure 1) mimics the body plan of a sub-carangiform fish (e.g. salmonids and koi), with a rigid head, flexible body and caudal fin. The head houses the control unit, battery and a balancing weight. The flexible body consists of three joints covered with waterproof rubber skin (as shown in figure 1*a*). The robotic fish is propelled by the undulation of the flexible body and the caudal fin (as shown in figure 1*b*). The caudal fin is also made of rubber but with stiffness decreasing towards the tip, mimicking the fin of a real fish [33,34]. Similar to real fish, our robot propels itself by body undulation (figure 1*b*).

### (b) The locomotion controller and kinematics

In addition to the body shape, the locomotion controller is also bioinspired. Real fish control their body undulation by a CPG neural network [35]. Consequently, we built an artificial CPG controller for our robotic fish. The controller can generate any undulation wave pattern with input of suitable parameters [31,36].

The controller model is as follows:

$$\dot{r}_i(t) = \alpha(R_i - r_i(t)), \tag{2.1}$$

$$\ddot{\phi} = \sum_{j=1}^{3} \mu[\mu(\phi_j(t) - \phi_i(t) - \varphi_{ij}) - 2(\dot{\phi}_i(t) - 2\pi f)] \tag{2.2}$$

and

$$\theta_i = r_i \cos(\phi_i(t)), \tag{2.3}$$

where $r_i$, $\phi_i$, $\theta_i$ are the state variables representing the undulation amplitude, phase and angle, respectively, of the *i*th joint. $R_i$ and $f$ determine the intrinsic amplitude scaled in degrees (°) and frequency is scaled in Hz. $\varphi_{ij}$ is the phase bias, which defines the phase offset between oscillators *i* and *j*. $\alpha$ and $\mu$ are the system parameters, which determine the system response speed. Both were set as 10 in our system, according to the dynamic response time of the servomotor.

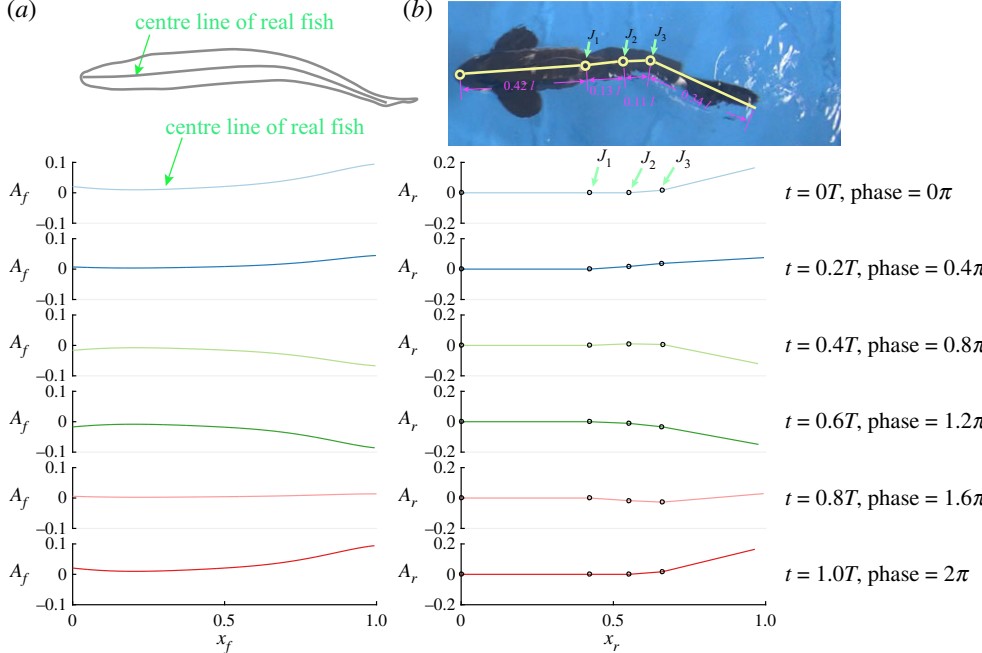

**Figure 2.** Body wave comparisons between real fish (saithe) and robotic fish. (*a*) Centre lines of real fish for a period of undulation at different time steps (plots are based on the data in fig. 4 in [37]). (*b*) Centre lines of robotic fish for a period of undulation mimicking a real fish's undulation pattern. $J_1$, $J_2$ and $J_3$ are the three joints of the robotic fish. $A_f$ and $A_r$ are, respectively, the amplitude of real fish and robotic fish. The amplitude and frequency of the robotic fish is double and half, respectively, of that of the real fish to keep the Strouhal number constant at 0.8. (Online version in colour.)

Based on the kinematic analysis of the undulation pattern of real fish (saithe) swimming at $0.86\,\mathrm{BL\,s^{-1}}$ (fig. 4 in [37]), we determined appropriate kinematics for our robot. However, limited by the power of the servomotors, our robot cannot reach very high-frequency undulations. To overcome this, we decreased the frequency but increased the amplitude to keep the Strouhal number constant at 0.8. The Strouhal number is a dimensionless number characterizing the arrangements of vortices in a wake [38,39]. It is defined as $fA/u$, where $f$ is the frequency, $A$ is the peak-to-peak amplitude and $u$ is the swimming speed. The parameters were set as $R_1 = 8°$, $R_2 = 12°$, $R_3 = 26°$, $f = 1\,\mathrm{Hz}$, $\varphi_{12} = 0.8164$, $\varphi_{13} = 1.6328$. The body waves of both real fish and robotic fish in one period are compared in figure 2. We also visualized the reverse von Kármán vortex generated by our robot by injecting dye into the water from a small pipe attached at the centre of the caudal fin (electronic supplementary material, figure S1). All these indicate that our robot generates a similar structure of vortex shedding to real fish under a similar Strouhal number.

## (c) The power acquisition module

In order to measure the energetic costs/benefits associated with hydrodynamic interactions between the robotic fish, we designed an accurate on-board power acquisition module (electronic supplementary material, figure S2). This consists of three main components: the voltage acquisition module, the current acquisition module and the data storage module. We added a data storage module to store the acquisition data locally because of the high sampling rate required to accurately measure the hydrodynamic influence. After each experiment, the data were sent to a computer for data analyses off-line.

To verify the effectiveness of our power acquisition module, we tested each swimming pattern five times and compared the results with those from the National Instrument (NI) acquisition (NI 9227 for the current acquisition and NI USB-6210 for the voltage acquisition). Figure 3a,b

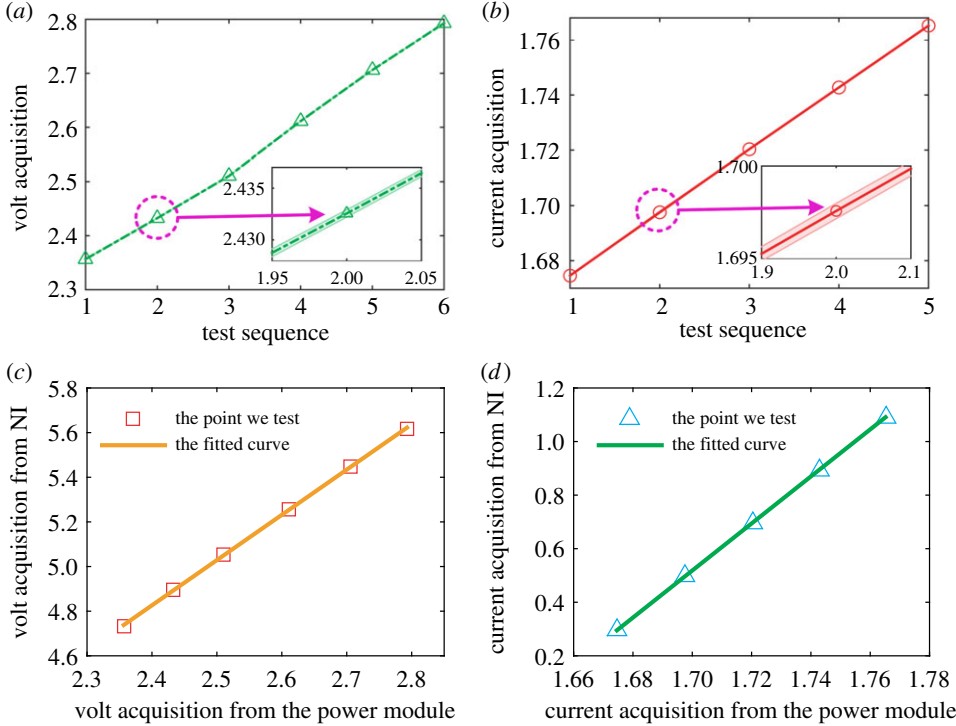

**Figure 3.** Verification of the power acquisition module. The volt acquisition (*a*) and current acquisition (*b*). (Inset) The enlargements around the second test point. (*c,d*) The relationship between our power acquisition module (volt (*a*) and current (*b*)) and the National Instrument (NI) power acquisition. (Online version in colour.)

illustrates the results of voltage and current detections for each swimming pattern. The shadow areas denote the standard error over the five tests. The insets in figure 3*a,b* show the enlargements around the second test point. All these indicate that the power acquisition module is stable.

To verify the effectiveness of the power acquisition system, we detected six different voltages and five different currents. The detected voltages and currents cover the range of the power costs of the robotic fish. Further calibrations were carried out by comparison with the NI acquisitions at the same voltage or current (figure 3*c,d*).

The calibration functions of the voltage and current module are deduced through the least squares method based on the test points (figure 3),

$$V = 2.027v - 0.0386 \tag{2.4}$$

and

$$C = 8.7819c - 14.4109, \tag{2.5}$$

where $v$ and $c$ are the values acquired from our power module and $V$ and $C$ are the real values calibrated by NI acquisition; the coefficients of determination ($R^2$) of these fittings are 0.9616 and 0.9979, respectively.

The power cost of the robot swimming alone for two periods of body undulation (tailbeats) is shown in figure 4*a*. Although the original power cost value is somewhat noisy, the trend of the signal is obvious after applying a Kalman filter (red curve in figure 4*a*). Since the speed of the tail flapping varies in one period (faster near the middle of the body central and almost zero when the tail reaches the maximum offset), we further analysed this signal by Fourier analysis (figure 4*b*). The first highest power spectral density (PSD) is at a frequency of 2 Hz, which is twice the tailbeat frequency. This is due to the symmetrical tail flapping in one period. The power cost of flapping from maximal leftmost move to rightmost equals that of the robot flapping from

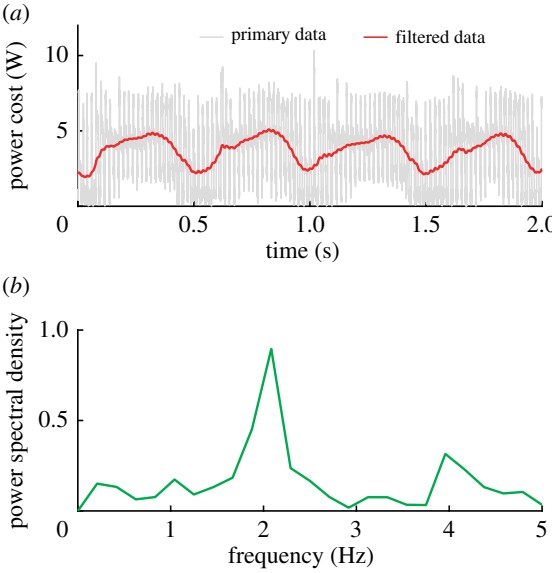

**Figure 4.** Verification of the energy cost. (*a*) Energy cost of one robot swimming freely by self-propulsion. (*b*) Spectral analysis of the energy cost by Fourier analysis. The first highest PSD is 2 Hz, which is twice the undulation frequency. (Online version in colour.)

maximal rightmost to leftmost. This demonstrates that our custom detection module was effective in measuring the power cost of our robot while swimming freely.

# 3. Methods and results

## (a) Experimental process

Experiments were carried out in a $3.0 \times 2.0 \times 0.4$ m tank. Benchmarking tests were performed where each robot first swam from one side to the other side (as shown in figure 5; see electronic supplementary material for details). The power costs and swim speeds were recorded as a baseline for comparing the power costs, speeds and efficiencies of robots swimming in groups. Speed evaluation was based on video captured from a global view camera positioned above a tank [40].

To evaluate the hydrodynamic interactions of swimming in formation, we developed a method to place two robotic fish side-by-side at 0.33 BL. The formation and distance have been suggested as a configuration that could allow real fish, and experimental physical models of fish-like swimming, to save energy when in a group [12,14,16,41]. Because of the turbulence and backflow, it is difficult to dynamically control and keep a stable formation between the two robots. To simplify this, we used a thin straight bar to maintain a stable formation between the two robots (as shown in figure 5*a*). Although the bar may transmit force between the robotic fish while limiting lateral movements, most forces were predominantly directed forwards (see electronic supplementary material, movie S1), indicating that the bar had only a very limited influence on straight forward swimming of the robots. Therefore, our system is also similar to previous simulations [13,41] or physical models [18], where lateral movements of the models were also restrained.

During the side-by-side swimming trials we initialized the two robotic fish with various relative undulation patterns, which we defined via the relative phase difference of their tailbeats. The definition of the phase for the robot is shown in figure 2. Phase difference was set by controlling the start-up time lag between the two robots. For example, two robots swimming

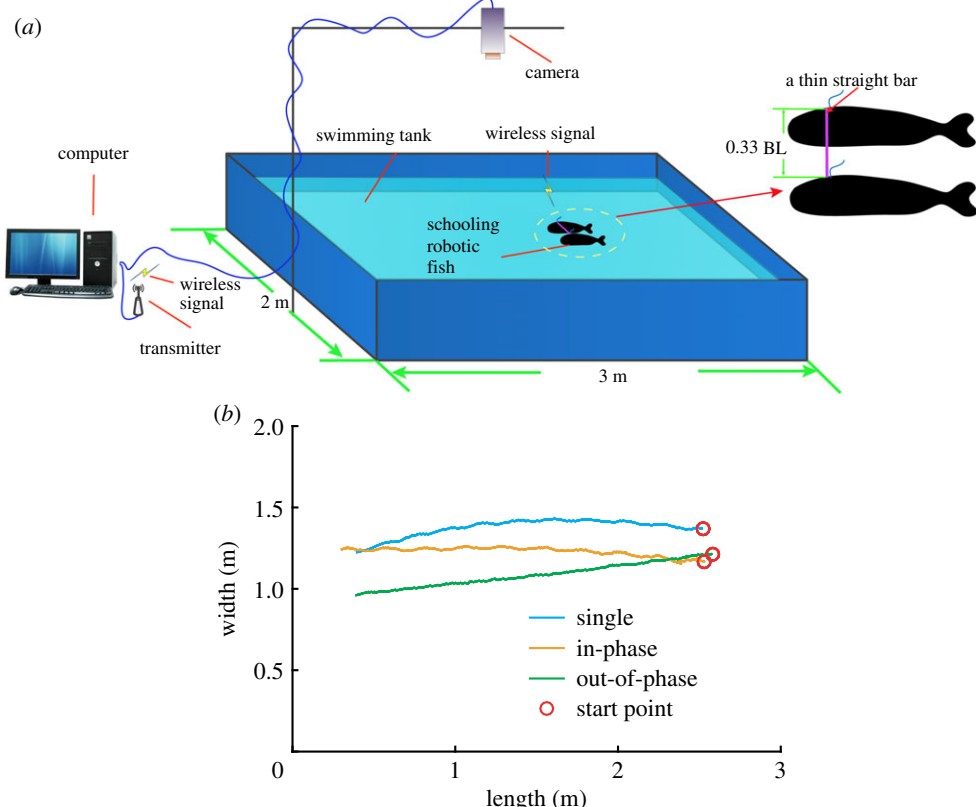

**Figure 5.** Schematic of the experimental set-up. (*a*) Illustration of the swim tank (3 m in length and 2 m in width) containing two robot fish connected by a thin bar that are controlled to swim from one side of the tank to the other under different undulation phase differences. (*b*) Sample trajectories of robotic fish swimming from one side of the tank to the other in isolation, as a pair swimming in-phase and as a pair swimming out-of-phase (trajectory of one robotic fish (centre point) is shown for the pair swimming as two robots swim side-by-side with a thin bar connected). (Online version in colour.)

out-of-phase (phase difference is $\pi$) was implemented by a half-undulation period delay (see electronic supplementary material, movie S1). Energy costs and swim speeds were recorded for each swimming pattern divided by those values recorded when swimming alone, resulting in power and speed coefficients (values greater than 1 representing greater speeds and power consumption, respectively) for each fish within the pair (see §3b for details).

Each was evaluated only once the robots had attained steady swimming (i.e. were no longer accelerating). In this way, we only analysed the swimming efficiency at uniform swim speed (by removing the starting/acceleration and stopping/deceleration segments of their trajectories). For each phase difference, we repeated each trial five times. Figure 5b illustrates the typical trajectories of robot fish swimming alone, in-phase and out-of-phase, respectively. They all moved forwards with little motion in the lateral direction. This indicates that the robots suffer limited lateral forces and thus the bar exerts only a very limited force on the robot. The oscillation in the lateral direction is maximal when the two robots swim in-phase and very small when swimming out-of-phase. This is due to the forces from the mediated water between two robotic fish cancelling each other when they swim out-of-phase.

## (b) Power costs with various hydrodynamic interactions

To evaluate the benefits or costs of hydrodynamic interactions between two swimming robots relative to swimming alone, we considered the power cost and swim speed relative to that of

swimming alone instead of the absolute power cost. We define the power coefficient $P_{coe}$ and speed coefficient $S_{coe}$ as

$$P_{coe} = \frac{P_s}{P_a} \tag{3.1}$$

and

$$S_{coe} = \frac{S_s}{S_a}, \tag{3.2}$$

where $P_a$ and $S_a$ are the power cost and swim speed of a robot swimming alone, respectively. $P_s$ and $S_s$ are the power cost and swim speed of robots swimming in groups, respectively. All the variables are scaled to the power and speed of the robot when swimming alone.

Following the power efficiency, analogous to 'miles per gallon' [32] or cost of transport [41], and considering all the times to be unit '1', we evaluate the efficiency as

$$\eta = \frac{S_{coe}}{P_{coe}}. \tag{3.3}$$

Therefore, the efficiency is 1 when the fish swims alone. Comparisons of power coefficient, speed coefficient and power efficiency are shown in figure 6. Statistical analyses employing ANOVA with post-hoc tests and Bonferroni corrections were conducted and are summarized in electronic supplementary material, figures S3 and S4.

Compared with swimming alone, our robot fish save energy for most phase differences (figure 6a; $p < 0.001$; first row in electronic supplementary material, figure S3); when swimming in pairs, they are both faster (figure 6b; $p < 0.001$; electronic supplementary material, figure S4) and more efficient (figure 6c) than when swimming alone, regardless of the phase relationship between them. Speed comparisons between the robots swimming alone and in groups with in-phase and out-of-phase patterns are shown in electronic supplementary material, movie S1.

The phase relationship between the robotic fish is found to substantially impact absolute speed and efficiency. Swimming exactly out-of-phase with their partner maximizes swim speed (figure 6b), but this comes at the cost of maximal average energetic expenditure of individuals in the pair (figure 6a). Conversely, the average energetic expenditure of both individuals in a pair is minimized when robotic fish exhibit body undulations that are exactly in-phase with one another. However, as shown in figure 6a, each individual can reduce, slightly ($p < 0.001$; electronic supplementary material, figure S3), their cost of locomotion if they adopt a phase difference of approximately $0.25\pi$, with respect to their neighbour. Doing so is a 'selfish' strategy that maximizes the power efficiency of the focal individual, but it has a substantial energetic cost to the focal individual's neighbour (figure 6a; $p < 0.001$; electronic supplementary material, figure S3).

## 4. Discussion

We demonstrate here that autonomous robotic fish, exhibiting similar body kinematics to real fish, can be an effective platform with which to investigate the biomechanics of swimming and the consequences of hydrodynamic interactions. Complementary to previous physical models employed to explore hydrodynamic interactions, our platform contributes in several ways. Firstly, compared with the majority of models that were designed to be fixed in place within moving flow [18,26,27,42], our robotic fish can swim freely, although with a thin bar connection for the side-by-side swimming conditions. We note that further improvements would be to include a force transducer in the bar to collect lateral and longitudinal forces or to attach the bar with hinges to each robot to allow for a longitudinal displacement while two robots are swimming side-by-side. Secondly, such a platform permits the *in situ* measurement of energy cost and swim speed instead of evaluating these indirectly through other detectable proxies [26,27]. Since the thrust and drag are both generated by the surrounding water, it is typically difficult to decouple the propulsion force from the resultant force [43]. Thirdly, most previous studies have considered Froude efficiency [18], which will be 0 when the robot swims at a constant speed [32,43]. We

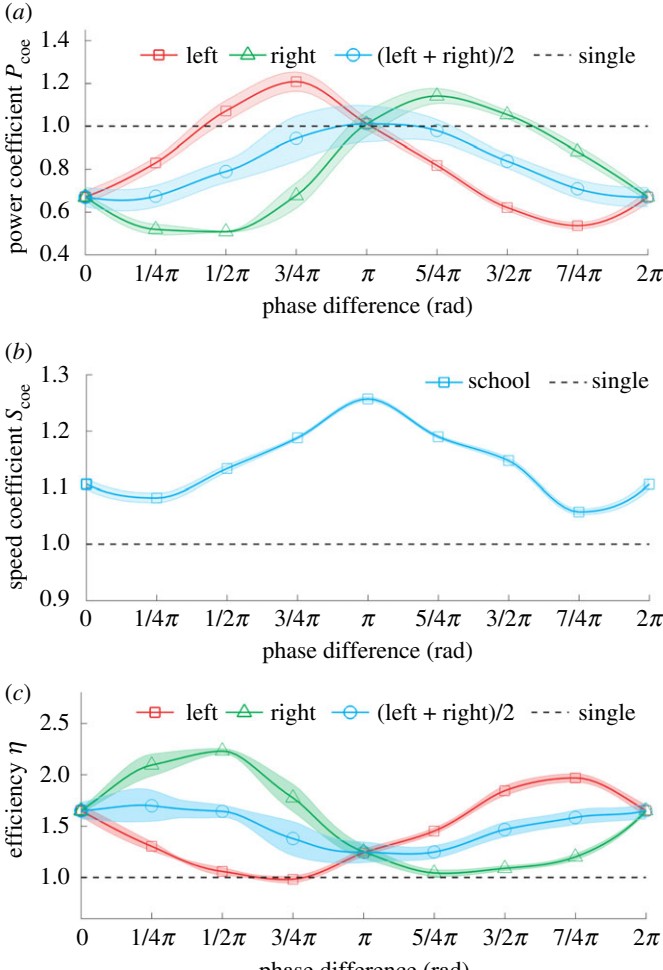

**Figure 6.** Locomotion performance of our robot when swimming in isolation and side-by-side with different undulation phase differences. The power coefficient (*a*), speed coefficient (*b*) and efficiency (*c*) all plotted as a function of the undulation phase difference. We performed statistical analysis by employing ANOVA with *post hoc* tests and Bonferroni corrections for the power costs between each two points in (*a*) (electronic supplementary material, figure S3) and swimming speeds between each two points in (*b*) (electronic supplementary material, figure S4). (Online version in colour.)

avoid this situation and define the efficiency as the speed divided by the power cost, following the idea of 'miles per gallon', as suggested by Schultz & Webb [32].

Our experiments demonstrate that, for biomimetic robots, swimming in pairs results in an increase in speed and efficiency, and a decrease in power consumption, when compared with swimming alone (figure 6). While this is similar to the results acquired from earlier studies [13,15,17,18,44], we additionally find that two fish swimming out-of-phase will probably benefit from a higher swimming efficiency, which was not previously observed [15,18] (figure 6). This novel finding is likely to be the result of the nominally realistic physical model we employ that accounts for the effects of three-dimensional body shape [45], self-propelled system [29] or flexible caudal fin [46]. The body platform is likely to contribute to the overall energetics since the three-dimensional shape would generate a vortex owing to undulation and the caudal fin may also extract energy from this vortex, such as described in [45]. The flexibility of the tail probably also contributes to the efficiency improvement in both individual and group swimming behaviour [46]. However, further experiments are required to assess the contribution of each of these factors.

There are several hydrodynamic mechanisms which may also contribute to higher efficiency when swimming side-by-side including the vortex interaction structure [17,44,47] and the channelling effect [13,47]. In our experiments, the trajectory of two fish swimming in-phase have larger oscillations in the lateral direction than when swimming out-of-phase (see electronic supplementary material, movie S1). This may be due to the formation of 'vortex-pair row' structures as described in [17], which result in a reduction in energetic cost and large lateral oscillations. Additionally, since the two robotic fish swim close together (0.33 BL), the speed of the flow between the robots will be increased owing to the channelling effect, resulting in a reduction in the power needed to generate thrust. This is similar to previous studies [13,47], where the channelling effect was found to be strong when two fish swim in similarly close proximity. Our model also includes a fish-like body, which will result in complex vortex interactions around the body that may also contribute to higher efficiency; for example, drag can be reduced by body–fin interactions [26,45,48] and by vortex interactions near the head [32]. Further experiments investigating the salient hydrodynamic mechanisms that result in energy saving and higher efficiency when fish swimming side-by-side will be useful.

Our results suggest that fish could, in a context-dependent way, balance the costs and benefits of swimming beside a conspecific by adjusting their relative phase relationship. This may provide a potential explanation for behavioural observations made on real fish, which indicate that they prefer to swim either in-phase, which would maximize efficiency of swimming at the cost of reduced speed, or out-of-phase, which would maximize speed at the cost of reduced efficiency, when swimming side-by-side in a flow tank [12,14]. Only for the in-phase and out-of-phase conditions are the speed and efficiency of the two robotic fish equal (figure 6). For other phase differences, the thrust may be slightly different between the two fish, resulting in a turning trajectory if they continued to maintain the side-by-side formation. Actively changing the phase difference, with respect to their neighbour, however, can allow individuals to minimize their energetic expenditure, but this proves to be a 'selfish' strategy that increases, considerably, the energetic cost experienced by their partner. This suggests the possibility of a game-theoretic dynamic within fish schools (which are typically composed of individuals of low relatedness), whereby individuals can exploit, and be exploited by, neighbours [49]. Our planned future studies will evaluate this possibility as well as consider hydrodynamic interactions of other swimming formations, with the objective of more coherently linking bio-inspiration to bio-understanding in collective motion.

Data accessibility. The data and codes used in this work are made available at: https://github.com/liatli/PRSA-Data-Codes.git.

Authors' contributions. L.L. and G.X. conceived the project. L.L. performed the experiments. L.L., S.R. and I.D.C. performed data analyses. S.R. and L.L. led the writing of the manuscript. I.D.C. edited the manuscript.

Competing interests. We declare we have no competing interests.

Funding. This work was supported in part by grants from the Natural Science Foundation of China (NSFC, grant no. 61633002) and the Beijing Natural Science Foundation (grant no. 4192026). I.D.C. acknowledges support from the NSF (grant no. IOS-1355061), the Office of Naval Research (ONR, grant no. N00014-19-1-2556), the Struktur-und Innovationsfunds für die Forschung of the State of Baden-Württemberg, the Deutsche Forschungsgemeinschaft (DFG, German Research Foundation) under Germany's Excellence Strategy–EXC 2117-422037984 and the Max Planck Society. S.R. was supported by the Alexander von Himboldt Foundation and acknowledges support from the Asian Office of Aerospace Research and Development (grant nos. FA23861914066 and FA23862014084). L.L. also acknowledges NVIDIA for a GPU Grant Program.

Acknowledgements. We thank Z. Pei for assistance with the experiments and the members of the G.X. and I.D.C. laboratories. We also thank J. Alex, K. Safi and D. Dechmann, for their discussion and suggestions.

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
