## [Peer Review File · Proceedings. Mathematical, Physical, and Engineering Sciences]

Review History

RSPA-2020-0810.R0 (Original submission)

Review form: Referee 1

Is the manuscript an original and important contribution to its field?

Good

Is the paper of sufficient general interest?

Good

Is the overall quality of the paper suitable?

Good

Can the paper be shortened without overall detriment to the main message?

Yes

Do you think some of the material would be more appropriate as an electronic appendix?

No

Do you have any ethical concerns with this paper?

No

Recommendation?

Major revision is needed (please make suggestions in comments)

Comments to the Author(s)

Review of the paper RSPA-2020-0810:

Using a robotic platform to study the influence of relative tailbeat phase on the energetic costs of side-by-side swimming in fish

By Li, Ravi, Xie, Couzin

This paper investigates the energetics of fish swimming in groups using a pair of biomimetic robotic fish. In the experiments the power consumed by the robots and their swimming speed are monitored, and the values for a single swimmer are used as comparison to quantify the effect of swimming with a neighbour. The parameter space controlling the problem comprises the frequency, amplitude and phase of the undulation of the two robotic swimmers. In this paper the tail beat amplitude and frequency are fixed and the focus is on the effect of the phase lag between the tail-beat kinematics of the two neighbouring fish. The paper is concisely written, describing the design of the experimental platform, the biomimetic approach for the locomotion controller and the swimming kinematics and the subtleties of the on-board power measurement. Ultimately, the discussion of the main result presented in figure 5 (the locomotion performance as probed by the changes in power consumption and swimming speed, and hence on efficiency, when a robot swims with a neighbour as compared to swimming alone) is convincing, which in my opinion warrants publication. I have nonetheless the following comments that should be addressed before.

Comments:

1) About section 2.2: The parameters of undulation are said to be "Based on kinematic analysis of the undulation pattern of real fish", with reference to Videler & Hess (1984), but no details are given, e.g. Is it the data from the Saithe or the Mackerel that is used as a model? Also, the Strouhal number is used to justify dynamical similarity of the robot with real fish so its definition should be recalled in the paper. What is its value compared to the fish used as model? Finally, in my opinion Figure 1 from the SI should be part of this section in the main paper, with more details given about the kinematics in physical units and in the dimensionless parameters. See also next comment on the vortex shedding figure that, if kept, should be part of this section 2.2.

2) About section 2.3. More suggestions about the structure of the paper: Figure 3 from the SI could be merged with figure 2 from the main paper to make a 4 panel figure, and the verification of power detection (section 2.4) together with Figure 4 from the SI should be part of section 2.3. This would make for a more coherent discussion of the on-board power detection system, which is one of the interesting technical advantages of the present study.

3) The part of section 2.4 that concerns the vortex shedding is visual but does not deserve a specific section, since it is just a qualitative comparison of the reverse von Kármán wake produced by the robot and that of a swimming trout from the literature. Unless a more thorough analysis of the evolution of the self-propelled wake, with specific cases of fish or model wakes at equivalent Strouhal numbers is performed, which is maybe out of the scope of the paper. As is I would suggest moving figure 3 to section 2.2, to accompany the discussion of dynamical similarity with fish.

4) The paragraph that reads "Speed evaluation was based on video captured from a global view camera positioned above a tank [37]. The robots were designed to swim from one side to the other side of the tank (see Section 3.1 for details)." in section 2.3 would be better placed in the methods section.

5) Concerning the side-by-side configuration experiments, please discuss the choice of the lateral distance between the two robots, which is an important parameter of the interaction. A useful reference for the change in Cost of Transport for a pair of swimmers as a function of the pattern of the "minimal school" is Li et al. "On the energetics and stability of a minimal fish school". Plos One 14(8), e0215265–20 (2019).

6) A final comment: An idea for future experiments would be to include a force transducer in the bar that maintains the side-by-side configuration, or to attach it with hinges to each robot to keep the side-by-side configuration but to allow for a longitudinal displacement that could find a stable position naturally (the front-back distance D examined in the authors' previous paper Li et al., Nat. Comms. 2020)

Minor comments:

- Some references are missing information. Please review. I noticed: Pages or Article number in Ref. [28], Volume, number etc in ref [14]. Also the authors' own Ref. [16] now has Volume, number etc information.

- Page 7: Typo: von Karmen -> von Kármán

- A couple of references to figures in section 1 of the SI are broken (Fig A. ??)

Review form: Referee 2

Is the manuscript an original and important contribution to its field?

Good

Is the paper of sufficient general interest?

Good

Is the overall quality of the paper suitable?

Excellent

Can the paper be shortened without overall detriment to the main message?

Yes

Do you think some of the material would be more appropriate as an electronic appendix?

No

Do you have any ethical concerns with this paper?

No

Recommendation?

Major revision is needed (please make suggestions in comments)

Comments to the Author(s)

This manuscript describes how pairs of bio-inspired robotic fish can save energy and/or increase speed as compared to when swimming alone. Whether speed or efficiency is maximized depends on the phase lag between the swimming robots. This is a very nice study that might be the starting point to many other studies that can try to evaluate the phase lag in real fish groups or build robotic fish swarms that swim most effectively. However, I have some issues that should be considered before publication:

(1) Are the found effects only valid with a fish shaped robot? I'm just curious to know whether

robots with slightly different shapes but a similar movement machinery would reach the same results. Is there anything known?

(2) I missed statistical analyses of the data that are above the level of pure descriptives. This is important to verify the found differences between phase differences as well as single and group tests.

(3) The authors state in the beginning that no data is shown in this paper ("My paper has no data") which is obviously not correct. So, please provide data and if possible the analysis code to enable other scientists to verify your results.

(4) Some parts of the figures are at low resolution, for example the little bar between the fish is almost invisible. Please check in the final figures to improve resolution.

Decision letter (RSPA-2020-0810.R0)

17-Feb-2021

Dear Dr Li

The Editor of Proceedings A has now received comments from referees on the above paper and would like you to revise it in accordance with their suggestions which can be found below (not including confidential reports to the Editor).

Please submit a copy of your revised paper within four weeks - if we do not hear from you within this time then it will be assumed that the paper has been withdrawn. In exceptional circumstances, extensions may be possible if agreed with the Editorial Office in advance.

Please note that it is the editorial policy of Proceedings A to offer authors one round of revision in which to address changes requested by referees. If the revisions are not considered satisfactory by the Editor, then the paper will be rejected, and not considered further for publication by the journal. In the event that the author chooses not to address a referee's comments, and no scientific justification is included in their cover letter for this omission, it is at the discretion of the Editor whether to continue considering the manuscript.

- Acknowledgements
- Funding statement

To revise your manuscript, log into <http://mc.manuscriptcentral.com/prsa> and enter your Author Centre, where you will find your manuscript title listed under "Manuscripts with Decisions." Under "Actions," click on "Create a Revision." Your manuscript number has been appended to denote a revision.

You will be unable to make your revisions on the originally submitted version of the manuscript. Instead, revise your manuscript and upload a new version through your Author Centre.

When submitting your revised manuscript, you will be able to respond to the comments made by the referee(s) and upload a file "Response to Referees" in Step 1: "View and Respond to Decision Letter". Please use this to document how you have responded to the comments, and the adjustments you have made. In order to expedite the processing of the revised manuscript, please be as specific as possible in your response to the referee(s).

IMPORTANT: Your original files are available to you when you upload your revised manuscript. Please delete any unnecessary previous files before uploading your revised version.

When revising your paper please ensure that it remains under 28 pages long. In addition, any pages over 20 will be subject to a charge (£150 + VAT (where applicable) per page). Your paper has been ESTIMATED to be 12 pages.

Once again, thank you for submitting your manuscript to Proc. R. Soc. A and I look forward to receiving your revision. If you have any questions at all, please do not hesitate to get in touch.

Yours sincerely
Raminder Shergill
proceedingsa@royalsociety.org

on behalf of
Dr Vikram Deshpande
Board Member
Proceedings A

Reviewer(s)' Comments to Author:
Referee: 1
Comments to the Author(s)
Review of the paper RSPA-2020-0810:

Using a robotic platform to study the influence of relative tailbeat phase on the energetic costs of side-by-side swimming in fish
By Li, Ravi, Xie, Couzin

This paper investigates the energetics of fish swimming in groups using a pair of biomimetic robotic fish. In the experiments the power consumed by the robots and their swimming speed are monitored, and the values for a single swimmer are used as comparison to quantify the effect of swimming with a neighbour. The parameter space controlling the problem comprises the frequency, amplitude and phase of the undulation of the two robotic swimmers. In this paper the tail beat amplitude and frequency are fixed and the focus is on the effect of the phase lag between the tail-beat kinematics of the two neighbouring fish. The paper is concisely written, describing the design of the experimental platform, the biomimetic approach for the locomotion controller and the swimming kinematics and the subtleties of the on-board power measurement. Ultimately, the discussion of the main result presented in figure 5 (the locomotion performance as probed by the changes in power consumption and swimming speed, and hence on efficiency, when a robot swims with a neighbour as compared to swimming alone) is convincing, which in my opinion warrants publication. I have nonetheless the following comments that should be addressed before.

Comments:

1) About section 2.2: The parameters of undulation are said to be "Based on kinematic analysis of the undulation pattern of real fish", with reference to Videler & Hess (1984), but no details are given, e.g. Is it the data from the Saithe or the Mackerel that is used as a model? Also, the Strouhal number is used to justify dynamical similarity of the robot with real fish so its definition should be recalled in the paper. What is its value compared to the fish used as model? Finally, in my opinion Figure 1 from the SI should be part of this section in the main paper, with more details given about the kinematics in physical units and in the dimensionless parameters. See also next comment on the vortex shedding figure that, if kept, should be part of this section 2.2.

2) About section 2.3. More suggestions about the structure of the paper: Figure 3 from the SI could be merged with figure 2 from the main paper to make a 4 panel figure, and the verification of power detection (section 2.4) together with Figure 4 from the SI should be part of section 2.3. This would make for a more coherent discussion of the on-board power detection system, which is one of the interesting technical advantages of the present study.

3) The part of section 2.4 that concerns the vortex shedding is visual but does not deserve a specific section, since it is just a qualitative comparison of the reverse von Kármán wake produced by the robot and that of a swimming trout from the literature. Unless a more thorough analysis of the evolution of the self-propelled wake, with specific cases of fish or model wakes at equivalent Strouhal numbers is performed, which is maybe out of the scope of the paper. As is I would suggest moving figure 3 to section 2.2, to accompany the discussion of dynamical similarity with fish.

4) The paragraph that reads "Speed evaluation was based on video captured from a global view camera positioned above a tank [37]. The robots were designed to swim from one side to the other side of the tank (see Section 3.1 for details)." in section 2.3 would be better placed in the methods section.

5) Concerning the side-by-side configuration experiments, please discuss the choice of the lateral distance between the two robots, which is an important parameter of the interaction. A useful reference for the change in Cost of Transport for a pair of swimmers as a function of the pattern of the "minimal school" is Li et al. "On the energetics and stability of a minimal fish school". Plos One 14(8), e0215265-20 (2019).

6) A final comment: An idea for future experiments would be to include a force transducer in the bar that maintains the side-by-side configuration, or to attach it with hinges to each robot to keep the side-by-side configuration but to allow for a longitudinal displacement that could found a stable position naturally (the front-back distance D examined in the authors' previous paper Li et al., Nat. Comms. 2020)

Minor comments:

- Some references are missing information. Please review. I noticed: Pages or Article number in Ref. [28], Volume, number etc in ref [14]. Also the authors' own Ref. [16] now has Volume, number etc information.

- Page 7: Typo: von Karmen -> von Kármán

- A couple of references to figures in section 1 of the SI are broken (Fig A. ??)

Referee: 2

Comments to the Author(s)

This manuscript describes how pairs of bio-inspired robotic fish can save energy and/or increase speed as compared to when swimming alone. Whether speed or efficiency is maximized depends on the phase lag between the swimming robots. This is a very nice study that might be the starting point to many other studies that can try to evaluate the phase lag in real fish groups or build robotic fish swarms that swim most effectively. However, I have some issues that should be considered before publication:

(1) Are the found effects only valid with a fish shaped robot? I'm just curious to know whether robots with slightly different shapes but a similar movement machinery would reach the same results. Is there anything known?

(2) I missed statistical analyses of the data that are above the level of pure descriptives. This is important to verify the found differences between phase differences as well as single and group tests.

(3) The authors state in the beginning that no data is shown in this paper (“My paper has no data”) which is obviously not correct. So, please provide data and if possible the analysis code to enable other scientists to verify your results.

(4) Some parts of the figures are at low resolution, for example the little bar between the fish is almost invisible. Please check in the final figures to improve resolution.

RSPA-2020-0810.R1 (Revision)

Review form: Referee 1

Is the manuscript an original and important contribution to its field?

Good

Is the paper of sufficient general interest?

Good

Is the overall quality of the paper suitable?

Good

Can the paper be shortened without overall detriment to the main message?

Yes

Do you think some of the material would be more appropriate as an electronic appendix?

No

Do you have any ethical concerns with this paper?

No

Recommendation?

Accept as is

Comments to the Author(s)

I thank the authors for their answers to my comments.

Review form: Referee 2

Is the manuscript an original and important contribution to its field?

Good

Is the paper of sufficient general interest?

Good

Is the overall quality of the paper suitable?

Acceptable

Can the paper be shortened without overall detriment to the main message?

Yes

Do you think some of the material would be more appropriate as an electronic appendix?

No

Do you have any ethical concerns with this paper?

Yes

Recommendation?

Accept with minor revision (please list in comments)

Comments to the Author(s)

The authors greatly improved the manuscript and dealt appropriately with most of the reviewers' comments. It might be a bit picky and I leave the decision with the editor whether or not to ask for a further revision but the included statistics (Figure 6 legend: "We performed a paired t-test for the power costs (Supplementary table 1) and swimming speeds (Supplementary table 2) under difference cases.") are not really satisfying. First, Figure 6a shows 4 different data sets (alone as a dashed line, right, left and (right+left)/2). How could 4 data sets compared with paired t-tests that are only able to compare 2 data sets. I suggest some kind of ANOVA with post-hoc tests in this case. Second, how could the 'alone' data set be compared with the others using paired t-tests when the alone data set is given without any variation. In such a case a one-sample t-test would work. Third, the results of the statistical tests are not mentioned in the main text (see above sentence which is the only reference to the tests) and any interpretation of the data is still done by visually evaluating the graphs. Fourth, by performing so many tests with the same data sets, some kind of adjustment to avoid p-value inflation like Bonferroni corrections is warranted. The data link is not assessable to me.

While I feel these issues are important to be fixed before publication, this is also easy to do and won't affect the data interpretation strongly.

Decision letter (RSPA-2020-0810.R1)

06-Apr-2021

Dear Dr Li,

On behalf of the Editor, I am pleased to inform you that your Manuscript RSPA-2020-0810.R1 entitled "Using a robotic platform to study the influence of relative tailbeat phase on the energetic costs of side-by-side swimming in fish" has been accepted for publication subject to minor revisions in Proceedings A. Please find the referees' comments below.

The reviewer(s) have recommended publication, but also suggest some minor revisions to your manuscript. Therefore, I invite you to respond to the reviewer(s)' comments and revise your manuscript. Please note that we have a strict upper limit of 28 pages for each paper. Please endeavour to incorporate any revisions while keeping the paper within journal limits. Please note that page charges are made on all papers longer than 20 pages. If you cannot pay these charges you must reduce your paper to 20 pages before submitting your revision. Your paper has been ESTIMATED to be 14 pages. We cannot proceed with typesetting your paper without your agreement to meet page charges in full should the paper exceed 20 pages when typeset. If you have any questions, please do get in touch.

It is a condition of publication that you submit the revised version of your manuscript within 7 days. If you do not think you will be able to meet this date please let me know in advance of the due date.

To revise your manuscript, log into <https://mc.manuscriptcentral.com/prsa> and enter your Author Centre, where you will find your manuscript title listed under "Manuscripts with

Decisions." Under "Actions," click on "Create a Revision." Your manuscript number has been appended to denote a revision.

You will be unable to make your revisions on the originally submitted version of the manuscript. Instead, revise your manuscript and upload a new version through your Author Centre.

When submitting your revised manuscript, you will be able to respond to the comments made by the referee(s) and upload a file "Response to Referees" in Step 1: "View and Respond to Decision Letter". You can use this to document any changes you make to the original manuscript. In order to expedite the processing of the revised manuscript, please be as specific as possible in your response to the referee(s).

IMPORTANT: Your original files are available to you when you upload your revised manuscript. Please delete any redundant files before completing the submission process.

When uploading your revised files, please make sure that you include the following as we cannot proceed without these:

- 1) A text file of the manuscript (doc, txt, rtf or tex), including the references, tables (including captions) and figure captions. Please remove any tracked changes from the text before submission. PDF files are not an accepted format for the "Main Document".
- 2) A separate electronic file of each figure (tif, eps or print-quality pdf preferred). The format should be produced directly from original creation package, or original software format.
- 3) Electronic Supplementary Material (ESM): all supplementary materials accompanying an accepted article will be treated as in their final form. Note that the Royal Society will not edit or typeset supplementary material and it will be hosted as provided. Please ensure that the supplementary material includes the paper details where possible (authors, article title, journal name). Supplementary files will be published alongside the paper on the journal website and posted on the online figshare repository (<https://figshare.com>). The heading and legend provided for each supplementary file during the submission process will be used to create the figshare page, so please ensure these are accurate and informative so that your files can be found in searches. Files on figshare will be made available approximately one week before the accompanying article so that the supplementary material can be attributed a unique DOI. Alternatively you may upload a zip folder containing all source files for your manuscript as described above with a PDF as your "Main Document". This should be the full paper as it appears when compiled from the individual files supplied in the zip folder.

Article Funder

Please ensure you fill in the Article Funder question on page 2 to ensure the correct data is collected for FundRef (<http://www.crossref.org/fundref/>).

Media summary

Please ensure you include a short non-technical summary (up to 100 words) of the key findings/importance of your paper. This will be used for to promote your work and marketing purposes (e.g. press releases). The summary should be prepared using the following guidelines:

*Write simple English: this is intended for the general public. Please explain any essential technical terms in a short and simple manner.

*Describe (a) the study (b) its key findings and (c) its implications.

*State why this work is newsworthy, be concise and do not overstate (true 'breakthroughs' are a rarity).

*Ensure that you include valid contact details for the lead author (institutional address, email address, telephone number).

Cover images

We welcome submissions of images for possible use on the cover of Proceedings A. Images should be square in dimension and please ensure that you obtain all relevant copyright permissions before submitting the image to us. If you would like to submit an image for consideration please send your image to proceedingsa@royalsociety.org

Once again, thank you for submitting your manuscript to Proceedings A and I look forward to receiving your revision. If you have any questions at all, please do not hesitate to get in touch.

Best wishes
 Raminder Shergill
proceedingsa@royalsociety.org
 Proceedings A

on behalf of
 Dr Vikram Deshpande
 Board Member
 Proceedings A

Reviewer(s)' Comments to Author:

Referee: 1
 Comments to the Author(s)
 I thank the authors for their answers to my comments.

Referee: 2
 Comments to the Author(s)
 The authors greatly improved the manuscript and dealt appropriately with most of the reviewers' comments. It might be a bit picky and I leave the decision with the editor whether or not to ask for a further revision but the included statistics (Figure 6 legend: "We performed a paired t-test for the power costs (Supplementary table 1) and swimming speeds (Supplementary table 2) under difference cases.") are not really satisfying. First, Figure 6a shows 4 different data sets (alone as a dashed line, right, left and (right+left)/2). How could 4 data sets compared with paired t-tests that are only able to compare 2 data sets. I suggest some kind of ANOVA with post-hoc tests in this case. Second, how could the 'alone' data set be compared with the others using paired t-tests when the alone data set is given without any variation. In such a case a one-sample t-test would work. Third, the results of the statistical tests are not mentioned in the main text (see above sentence which is the only reference to the tests) and any interpretation of the data is still done by visually evaluating the graphs. Fourth, by performing so many tests with the same data sets, some kind of adjustment to avoid p-value inflation like Bonferroni corrections is warranted. The data link is not assessable to me.
 While I feel these issues are important to be fixed before publication, this is also easy to do and won't affect the data interpretation strongly.

Decision letter (RSPA-2020-0810.R2)

06-Apr-2021

Dear Dr Li,

On behalf of the Editor, I am pleased to inform you that your Manuscript RSPA-2020-0810.R1 entitled "Using a robotic platform to study the influence of relative tailbeat phase on the energetic costs of side-by-side swimming in fish" has been accepted for publication subject to minor revisions in Proceedings A. Please find the referees' comments below.

The reviewer(s) have recommended publication, but also suggest some minor revisions to your manuscript. Therefore, I invite you to respond to the reviewer(s)' comments and revise your manuscript. Please note that we have a strict upper limit of 28 pages for each paper. Please endeavour to incorporate any revisions while keeping the paper within journal limits. Please note that page charges are made on all papers longer than 20 pages. If you cannot pay these charges you must reduce your paper to 20 pages before submitting your revision. Your paper has been ESTIMATED to be 14 pages. We cannot proceed with typesetting your paper without your agreement to meet page charges in full should the paper exceed 20 pages when typeset. If you have any questions, please do get in touch.

It is a condition of publication that you submit the revised version of your manuscript within 7 days. If you do not think you will be able to meet this date please let me know in advance of the due date.

To revise your manuscript, log into <https://mc.manuscriptcentral.com/prsa> and enter your Author Centre, where you will find your manuscript title listed under "Manuscripts with Decisions." Under "Actions," click on "Create a Revision." Your manuscript number has been appended to denote a revision.

You will be unable to make your revisions on the originally submitted version of the manuscript. Instead, revise your manuscript and upload a new version through your Author Centre.

When submitting your revised manuscript, you will be able to respond to the comments made by the referee(s) and upload a file "Response to Referees" in Step 1: "View and Respond to Decision Letter". You can use this to document any changes you make to the original manuscript. In order to expedite the processing of the revised manuscript, please be as specific as possible in your response to the referee(s).

IMPORTANT: Your original files are available to you when you upload your revised manuscript. Please delete any redundant files before completing the submission process.

When uploading your revised files, please make sure that you include the following as we cannot proceed without these:

- 1) A text file of the manuscript (doc, txt, rtf or tex), including the references, tables (including captions) and figure captions. Please remove any tracked changes from the text before submission. PDF files are not an accepted format for the "Main Document".
- 2) A separate electronic file of each figure (tif, eps or print-quality pdf preferred). The format should be produced directly from original creation package, or original software format.
- 3) Electronic Supplementary Material (ESM): all supplementary materials accompanying an accepted article will be treated as in their final form. Note that the Royal Society will not edit or typeset supplementary material and it will be hosted as provided. Please ensure that the

supplementary material includes the paper details where possible (authors, article title, journal name). Supplementary files will be published alongside the paper on the journal website and posted on the online figshare repository (<https://figshare.com>). The heading and legend provided for each supplementary file during the submission process will be used to create the figshare page, so please ensure these are accurate and informative so that your files can be found in searches. Files on figshare will be made available approximately one week before the accompanying article so that the supplementary material can be attributed a unique DOI. Alternatively you may upload a zip folder containing all source files for your manuscript as described above with a PDF as your "Main Document". This should be the full paper as it appears when compiled from the individual files supplied in the zip folder.

Article Funder

Please ensure you fill in the Article Funder question on page 2 to ensure the correct data is collected for FundRef (<http://www.crossref.org/fundref/>).

Media summary

Please ensure you include a short non-technical summary (up to 100 words) of the key findings/importance of your paper. This will be used for to promote your work and marketing purposes (e.g. press releases). The summary should be prepared using the following guidelines:

*Write simple English: this is intended for the general public. Please explain any essential technical terms in a short and simple manner.

*Describe (a) the study (b) its key findings and (c) its implications.

*State why this work is newsworthy, be concise and do not overstate (true 'breakthroughs' are a rarity).

*Ensure that you include valid contact details for the lead author (institutional address, email address, telephone number).

Cover images

We welcome submissions of images for possible use on the cover of Proceedings A. Images should be square in dimension and please ensure that you obtain all relevant copyright permissions before submitting the image to us. If you would like to submit an image for consideration please send your image to proceedingsa@royalsociety.org

Once again, thank you for submitting your manuscript to Proceedings A and I look forward to receiving your revision. If you have any questions at all, please do not hesitate to get in touch.

Best wishes

Raminder Shergill

proceedingsa@royalsociety.org

Proceedings A

on behalf of

Dr Vikram Deshpande

Board Member

Proceedings A

Reviewer(s)' Comments to Author:

Referee: 1

Comments to the Author(s)

I thank the authors for their answers to my comments.

Referee: 2

Comments to the Author(s)

The authors greatly improved the manuscript and dealt appropriately with most of the reviewers' comments. It might be a bit picky and I leave the decision with the editor whether or not to ask for a further revision but the included statistics (Figure 6 legend: "We performed a paired t-test for the power costs (Supplementary table 1) and swimming speeds (Supplementary table 2) under difference cases.") are not really satisfying. First, Figure 6a shows 4 different data sets (alone as a dashed line, right, left and $(\text{right}+\text{left})/2$). How could 4 data sets compared with paired t-tests that are only able to compare 2 data sets. I suggest some kind of ANOVA with post-hoc tests in this case. Second, how could the 'alone' data set be compared with the others using paired t-tests when the alone data set is given without any variation. In such a case a one-sample t-test would work. Third, the results of the statistical tests are not mentioned in the main text (see above sentence which is the only reference to the tests) and any interpretation of the data is still done by visually evaluating the graphs. Fourth, by performing so many tests with the same data sets, some kind of adjustment to avoid p-value inflation like Bonferroni corrections is warranted. The data link is not assessable to me.

While I feel these issues are important to be fixed before publication, this is also easy to do and won't affect the data interpretation strongly.